# Mechanical Study of Various Pedicle Screw Systems including Percutaneous Pedicle Screw in Trauma Treatment

**DOI:** 10.3390/medicina58050565

**Published:** 2022-04-20

**Authors:** Yoshiaki Oda, Tomoyuki Takigawa, Yasuo Ito, Haruo Misawa, Tomoko Tetsunaga, Koji Uotani, Toshifumi Ozaki

**Affiliations:** 1Department of Orthopaedic Surgery, Okayama University Hospital, 2-5-1 Shikata-cho, Kitaku, Okayama City 700-8558, Japan; orthosec@md.okayama-u.ac.jp (H.M.); kwdtmk1201@yahoo.co.jp (T.T.); coji.uo@gmail.com (K.U.); 2Department of Orthopaedic Surgery, Kobe Red Cross Hospital, 1-3-1 Wakinohamakaigandori, Chuoku, Kobe City 651-0073, Japan; m-izutsu@kobe.jrc.or.jp (T.T.); y-ito@kobe.jrc.or.jp (Y.I.); 3Department of Orthopaedic Surgery, Dentistry, and Pharmaceutical Sciences, Graduate School of Medicine, Okayama University, 2-5-1 Shikata-cho, Kitaku, Okayama City 700-8558, Japan; tozaki@md.okayama-u.ac.jp

**Keywords:** spine surgery, percutaneous pedicle screw, percutaneous systems, break test, fatigue test, biomechanical study discipline

## Abstract

*Background and Objectives*: Spine surgery using a percutaneous pedicle screw placement (PPSP) is widely implemented for spinal trauma. However, percutaneous systems have been reported to have weak screw–rod connections. In this study, conventional open and percutaneous systems were biomechanically evaluated and compared. *Material and Methods*: The experiments were performed in two stages: the first stage was a break test, whereas the second stage was a fatigue test. Four systems were used for the experiments. System 1 was intended for conventional open surgery (titanium rod with a 6.0 mm diameter, using a clamp connecting mechanism). System 2 was a percutaneous pedicle screw (PPS) system for trauma (titanium alloy rod with a 6.0 mm diameter, using ball ring connections). System 3 was a PPS system for trauma (cobalt–chromium alloy rod with a 6.0 mm diameter, using sagittal adjusting screw connections). System 4 was a general-purpose PPS system (titanium alloy rod with a 5.5 mm diameter, using a mechanism where the adapter in the head holds down the screw). *Results*: Stiffness values of 54.8 N/mm, 43.1 N/mm, 90.9 N/mm, and 39.3 N/mm were reported for systems 1, 2, 3, and 4, respectively. The average number of load cycles in the fatigue test was 134,393, 40,980, 1,550,389, and 147,724 for systems 1 to 4, respectively. At the end of the test, the displacements were 0.2 mm, 16.9 mm, 1.2 mm, and 8.6 mm, respectively. System 1, with a locking mechanism, showed the least displacement at the end of the test. *Conclusion*: A few PPS systems showed better results in terms on stiffness and life than the open system. The experiments showed that mechanical strength varies depending on the spinal implant. The experiments conducted are essential and significant to provide the mechanical strength required for surgical reconstruction.

## 1. Introduction

Technological advances have enabled the application of minimally invasive surgical techniques in spinal surgery. The percutaneous pedicle screw placement (PPSP) is a widely used minimal spinal surgical technique. Moreover, PPS has been applied to various pathologies [1,2,3,4,5,6,7,8,9,10,11]. When performing PPS, the surgeon can select the rod material and diameter. Furthermore, various systems for spinal trauma enabling percutaneous fracture reduction have been developed, which differ from the mobile polyaxial screw between the screw and screw head. The spinal implant can be selected based on the patient’s pathology and planned surgery. However, hooks, a sublaminar taping, or a crosslink cannot be added to the posterior fixation by PPS. In particular, posterior fixation by PPS could be weaker than posterior fixation by conventional systems. Therefore, this study compares the biomechanics of spinal implant systems used for trauma. This study is the first to investigate the mechanical property from traditional trauma systems to recent percutaneous systems.

## 2. Methods

The experiment was conducted in two stages. The first stage was a break test. The second stage was a fatigue test. The purpose of the break test was to examine the stiffness, yield values, and failure of the spinal implant systems and measure the stress of the rods to set the load for the fatigue test in the second stage. The purpose of the fatigue test was to investigate the fatigue life; particularly, to observe the amount of displacement in one load cycle and permanent deformation seen at the end of the test. The experimental apparatus used was an Instron 8521 (Instron 8521, Norwood, MA, USA). The experiments were conducted according to the ASTM 1717-13 [12] using a vertebrectomy model. Thus, the spinal implant system was installed in an ultra-high molecular weight polyethylene block (Figure 1).

Four different spinal systems were used in the experiments (Figure 2).

System 1—open system for trauma, screw diameter: 6.2 mm, screw material: Ti-6Al-7Nb, rod diameter: 6.0 mm, rod material: titanium (Ti), connection mechanism: fixing system using clamps.System 2—PPS system for trauma, screw diameter: 6.5 mm, screw material: Ti-6Al-4V, rod diameter: 6.0 mm, rod material: Ti-6Al-4V, connecting mechanism: fastening using a ball ring in the offset connector.System 3—PPS system for trauma, screw diameter: 6.5 mm, screw material: screw shaft of Ti-6Al-4V, screw head of cobalt–chrome alloy (not disclosed), rod diameter: 6.0 mm, rod material: cobalt–chromium alloy (not disclosed), connecting mechanism: using a sagittal adjusting screw.System 4—general PPS system, screw diameter: 6.5 mm, rod material: Ti-6Al-4V, rod diameter: 5.5 mm, rod material: Ti-6Al-4V, connecting mechanism: a mechanism where the adapter inside the screw head holds down the screw head.

### 2.1. Break Test

The experiments were conducted using one set of four spinal systems. An eccentric load was applied to the rods at increments of 3.7 N/s, and the load was stopped when an evident plastic deformation was observed. The yield value (N) represents the force at the point where a displacement of 2% of the effective rod length. That is, a displacement of 1.5 mm is observed, as defined by ASTM F1717. Stiffness was calculated as a tangent of the initial slope of the load-displacement curve. The initial slope up to 2% displacement was defined as stiffness. A strain gauge was installed on the rod to measure the strain on the rod (Figure 3). The Young’s modulus of titanium was defined as 110 GPa, and the Young’s modulus of cobalt–chromium was 210 GPa. The stress of the rod was calculated from the measured strain of the rod. The load of the fatigue test corresponded to the value at which the initial failure did not occur while increasing the internal stress of the rod obtained by calculation as much as possible.

### 2.2. Fatigue Test

A cyclic axial compression load accompanied by bending was applied. The maximum load was defined based on the result of the break test and the fatigue limit of 10 million times for Ti-6Al-4V at 740 MPa [13]. The constant load ratio was 10, and the load was repeated at 1 Hz until the spinal implant assembly broke or up to 2,000,000 cycles. We performed three sets for four spinal systems. The measured items were the number of cycles until the end of the test, the displacement (mm) of the spinal implant assembly observed during one cycle (observed at the first load, midpoint, and final load), and the displacement (mm) of the spinal implant assembly at the end of the test. An example is presented in Figure 4.

## 3. Results

Table 1 and Figure 5 show the results of the break tests. The yield values (N) are 79.3 N, 64.6 N, 136.8 N, and 57.0 N for System 1, System 2, System 3, and System 4, respectively. The stiffness values (N/mm) for System 1, System 2, System 3, and System 4 are 54.8 N/mm, 43.1 N/mm, 90.9 N/mm, and 39.3 N/m, respectively. The failure occurred in System 2 above 400 N. The external force–strain curve of System 4, with titanium alloy (Ti-6Al-4V) and the thinnest rod diameter of 5.5 mm, is shown in Figure 6. The calculated rod stress under 400 N external force was in the range of 666–700.0 MPa Therefore, 400 N was considered as an appropriate value for the fatigue test. Under an external force of 400 N, the stress of the rod approached the fatigue limit, and the maximum load of the fatigue test was set to 400 N.

The fatigue test results are listed in Table 2. The average number of cycles at the end of the test was 134,393, 40,980, 1,550,389, and 147,724 for systems 1 to 4, respectively (Figure 7). All assemblies corresponding to systems 1, 2, and 4 were broken, and only one assembly was broken in System 3. All System 1 assemblies were broken at the base of the screw, and all System 4 assemblies were broken at the rod.

The displacement amount (mm) during one cycle at the time of the first load was 7.6 mm, 13.7 mm, 4.8 mm, and 2.4 mm, for systems 1 to 4, respectively. The displacement amount (mm) at the midpoint in the fatigue test was 8.8 mm, 11.7 mm, 4.4 mm, 9.6 mm, respectively. The displacements (mm) at the final load were 9.2, 11.9, 4.4, and 9.0, respectively. The displacements (mm) at the end of the test were 0.2 mm, 16.9 mm, 1.2 mm, and 8.6 mm, respectively (Figure 8).

## 4. Discussion

Posterior fixation is a typical procedure for thoracolumbar fractures. Since the introduction of the PPS, posterior fixation with PPS has become widespread in trauma treatment. In fracture treatment, maintaining the correction until completing bone fusion is essential. The connecting parts, rods, and interfaces between the screws and the bone are prone to dislocation due to bending, deformation, and loosening. However, polyaxial PPS might have low stability and maintain fracture reduction. Correction loss in the sagittal plane has been reported [14]. Moreover, Fogel et al. [15] reported that failures occur between the screw head and screw shafts with polyaxial screws. Therefore, systems with a locking mechanism to avoid correction loss are recommended [16]. Since the introduction of the PPS, their clinical application has progressed, and various improvements have been obtained, focusing on the connecting part between the screw head and the screw shaft. The characteristics of each system were presented in this study. The following spinal implant systems for spinal trauma were considered and mechanically tested in this study: System 1 is used in conventional open surgical procedures, Systems 2 and 3 are PPS systems, and System 4 is a general-purpose PPS system. There are few reports of studies similar to this experiment; thus, this study significantly contributes to current clinical practice. When there is a concern about initial failure (such as when AO classification A3 or higher with large angular deformation or when the large surgical correction is maintained only with the PPS system), the conventional multiaxial screw should be avoided. The results of this study show that a system with a connecting mechanism with angular stability, or a multiaxial screw system that has been improved so that initial failure is less likely to occur should be used.

In this study, the vertebral body model used a high molecular weight polyethylene block; thus, almost no movement existed between the polyethylene block and the screw. Therefore, the displacement (mm) can be assumed as the sum of the deformation of the screw shaft, the connecting part, and the rod. Bone quality is crucial in clinical practice. With good bone quality, the interface between the bone and the screw is unlikely to loosen; thus, it is considered possible to maintain the spine with greater force. To achieve this, it may be appropriate to select a thick rod system with high yield point and high stiffness. Using a thick rod will extend the life of the implant and may be advantageous even if it takes time to heal the fracture. On the contrary, if the bone quality is poor, the corrective maintenance of the spine by the spinal implant system (PPS) is limited, and it is considered that there is no merit in selecting an implant system with a high yield point and stiffness. When performing major surgical corrections for patients with poor bone quality, we consider that anterior reconstruction and vertebroplasty should be performed by increasing the number of screws to prevent large forces from concentrating on the spine and implants.

System 1 is a clamp-based connection mechanism previously recommended for use in trauma. The system has a structure that does not easily loosen in the same direction. The final displacement (mm) of System 1 was the smallest among the four spinal systems, probably because of the connection mechanism by the clamp. Even in System 3.4 without a locking mechanism, the dislocation at the end of the test was less than 2 mm, which was less than expected. System 1 has a connecting mechanism with clamps and has a locking mechanism structure that does not easily loosen in the same direction. The results show that the final displacement (mm) of System 1 was the smallest of the four spinal systems. The connecting mechanism of System 2 is not a locking mechanism; instead, it is a structure where a screw puts pressure on a spherical surface. The early failure observed in the break test was consistent with the concern that the biomechanical stability of the polyaxial screw was low. In addition, the amount of displacement (mm) observed during one cycle of the fatigue test and the amount of the final displacement at the end of the fatigue test were the largest of the four spinal systems. The connecting mechanism might be the cause of the large amount of displacement in System 2. System 3 was not a polyaxial screw but a mechanism where no movement was observed between the screw and screw head. System 3 has a different mechanism from conventional polyaxial and does not move between the screw and the screw head. System 3 is a system that could also use a titanium alloy rod. However, a cobalt–chrome alloy rod with a 6.0 mm diameter was used in this test. The amount of displacement during one cycle of the fatigue test was the smallest, and the final displacement after the fatigue test was 1.2 mm. The connection mechanism of System 4 has an inner blocker installed in the screw head, and the inner block grips the rod when the set screw is fastened. System 4 is a general-purpose PPS system that is not intended for trauma. The biomechanical stability of the connecting part was expected to be low; however, the displacement during the fatigue test was less than expected. Even with polyaxial screws, systems with improved connecting mechanisms may have improved the biomechanical weaknesses demonstrated in the past.

The stiffness of the system is also essential for maintaining the correction until complete bone fusion. In the break test, the cobalt–chromium alloy rod with a 6.0 mm diameter had the highest stiffness. This system was followed by System 1 of Φ6.0 mm titanium, System 2 of Φ6.0 mm titanium alloy, and System 4 of Φ5.5 mm titanium alloy. These are the values calculated from the initial slope. In particular, in this study, the stiffness of System 2 was higher than that of System 4, but failure occurred when the external force exceeded 400 N in the break test.

Rohlmann et al. [17] measured the force applied to a spinal implant in vivo. The study reported that the force exerted on the spinal implant during walking was slightly below 400 N. The load on the spinal implant increased caudally. It is considered that the maximum load of the 400 N set in this fatigue test was an adequate value. When only posterior fixation by PPS for an unstable thoracolumbar fracture and early standing and walking training after surgery is performed, the unstable fractured vertebral body may resemble this fatigue test. As seen in this experiment, the bone union is unlikely to occur in situations where 8–10 mm movement occurs at the fracture. Reconstruction of the anterior column should be required for more reliable bone fusion and early ambulation. Noshchenko et al. [18], Kubosch et al. [19], and Demura et al. [20] reported that cobalt–chromium alloy rods had higher bending stiffness than titanium alloy rods. Similar results were obtained in this study.

In the fatigue test, System 3 using Φ6.0 mm cobalt–chrome alloy had the longest life, followed by System 1 and System 3, which had a similar lifespan. In a fatigue test, Nguyen et al. [21] reported an average lifespan of 350,000 times longer than that of titanium alloys at loads up to 400 N. This study did not consider changing only the rod material. However, this aspect did not contradict that the system using cobalt–chromium alloy had a longer life than the system using titanium alloy rods. System 4, using a Φ5.5 mm titanium alloy rod without a locking mechanism, had a life similar to System 1. This result was the opposite of what we expected.

All ruptures of System 1 during fatigue testing were seen at the base of the screw. In System 4 with the titanium alloy rod of Φ5.5 mm, all ruptures were observed in the rod. System 2 ruptures were seen on both screws and rods. There was a certain tendency depending on the spinal system. If the screw remains horizontal, the shear force may be exerting a large amount on the screw. If a load is repeatedly applied at a position that is largely displaced from the initial position, the shearing force applied to the screw will decrease, and there is a possibility that a large bending stress will be applied. In an environment where a large vertical force is applied to the screw, as in this experiment, the rod and screw diameters and the design could help improve the life of the implant.

The limitation of this experiment is that no torsion or lateral bending tests were performed. Because the experiment uses blocks of polyethylene having a high molecular weight, the screws and the vertebral body were not loosened. If the vertebral bone is substantially stiff so that the screws will not loosen, the results of this experiment will be useful. However, in cases such as osteoporosis, loosening between the vertebral body and the screw is also a crucial factor. This is an aspect to study in the future. The results of this experiment could not draw statistical conclusions because the sample size is extremely small. Break tests were conducted on only one sample of each system, and results were very limited. A sample size that ensures sufficient power even in a fatigue test was not used; therefore, it is difficult to draw conclusions based on the results of this study.

## 5. Conclusions

We compared the conventional open spinal implant system and the PPS spinal implant using a biomechanical test. The weakness of the connection mechanism of the PPS system, reported previously, improved depending on the spinal system. The amount of displacement with respect to the repetitive load was larger for the titanium rod than for the cobalt–chrome rod, and the fatigue life was longer for the cobalt–chrome rod.

## Figures and Tables

**Figure 1 medicina-58-00565-f001:**
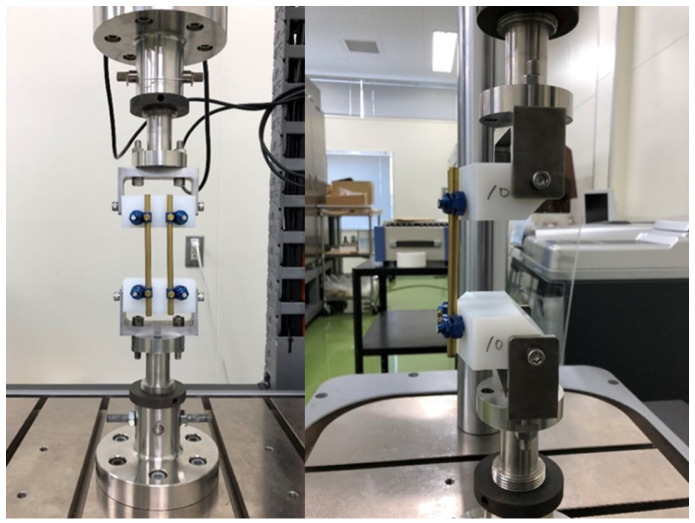
Test specimen installed on the Instron 8521.

**Figure 2 medicina-58-00565-f002:**
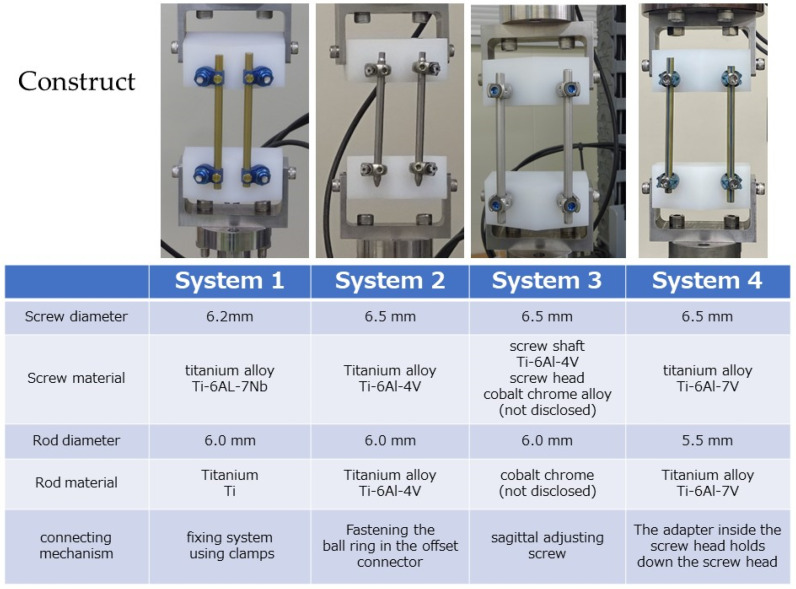
Data of the four spinal systems used in this experiment.

**Figure 3 medicina-58-00565-f003:**
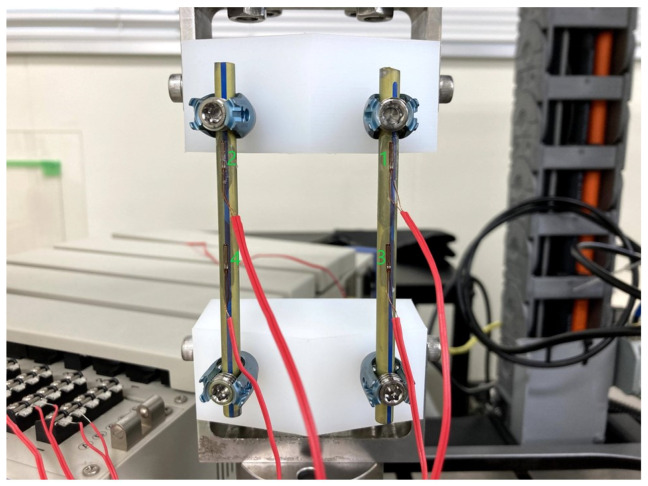
Four strain gauges were installed on the rod to measure the strain in the rod. The numbers are shown for channel identification.

**Figure 4 medicina-58-00565-f004:**
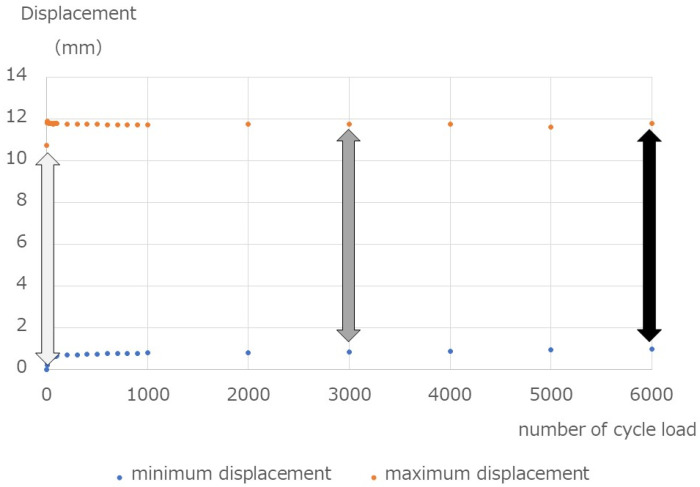
Displacements during the fatigue test. The specimen was repeatedly loaded with 40 N–400 N, and the minimum and maximum displacements observed in one cycle were recorded. The three arrows indicate that the difference was measured at three points: the first load (light gray arrow), midpoint in the fatigue test (dark gray arrow), and final load (black arrow). Moreover, the displacement after the test was recorded.

**Figure 5 medicina-58-00565-f005:**
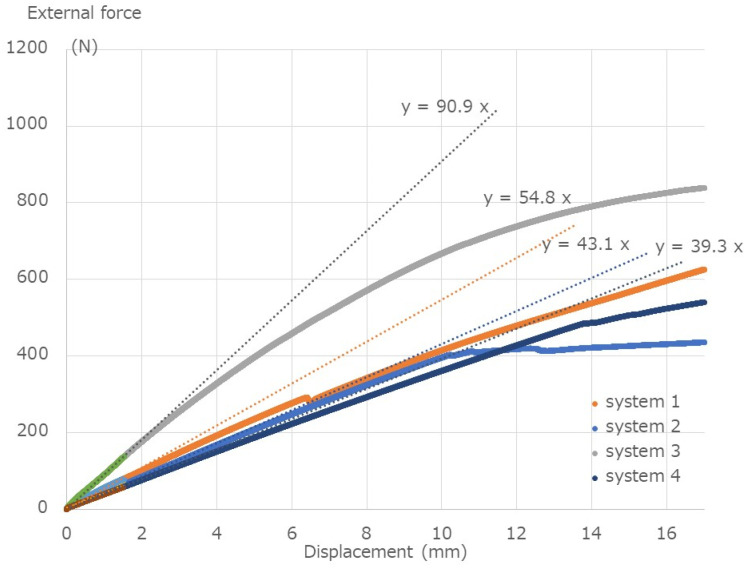
Break test results. The external force–displacement curve shows that the initial slopes are Y = 54.8 × X for System 1, Y = 43.1 × X for System 2, Y = 90.9 × X for System 3, and Y = 39.3 × X for System 4.

**Figure 6 medicina-58-00565-f006:**
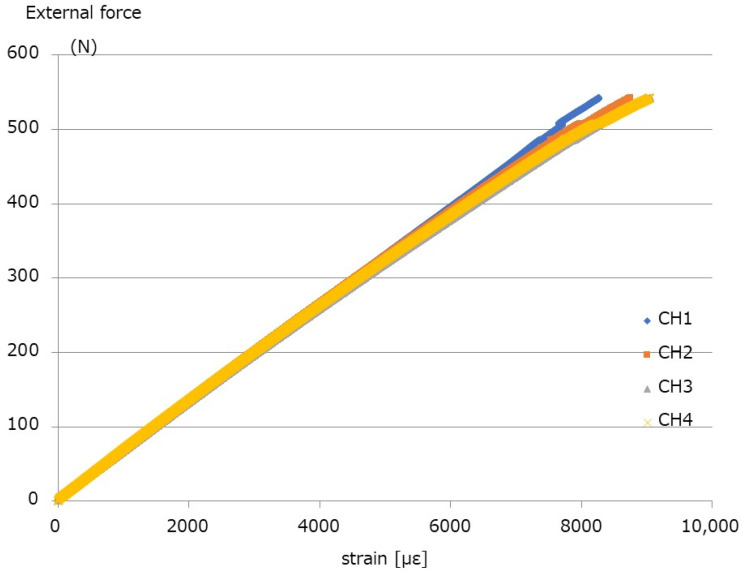
External force–strain curve of System 4. The CH number corresponds to the number shown in Figure 2.

**Figure 7 medicina-58-00565-f007:**
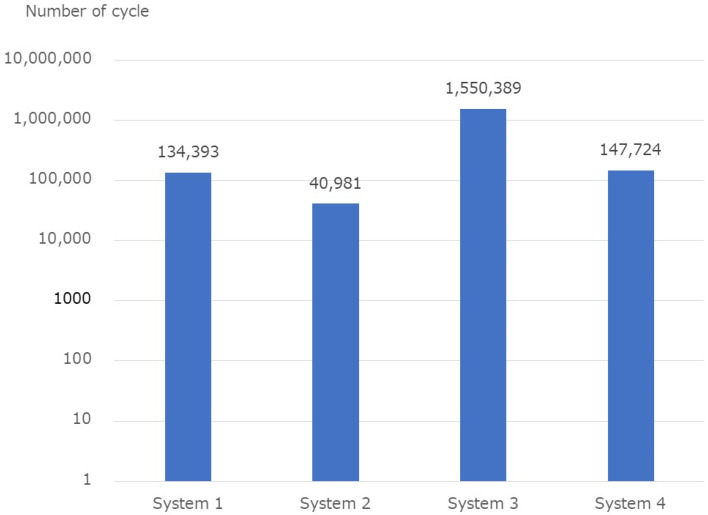
Fatigue test results.

**Figure 8 medicina-58-00565-f008:**
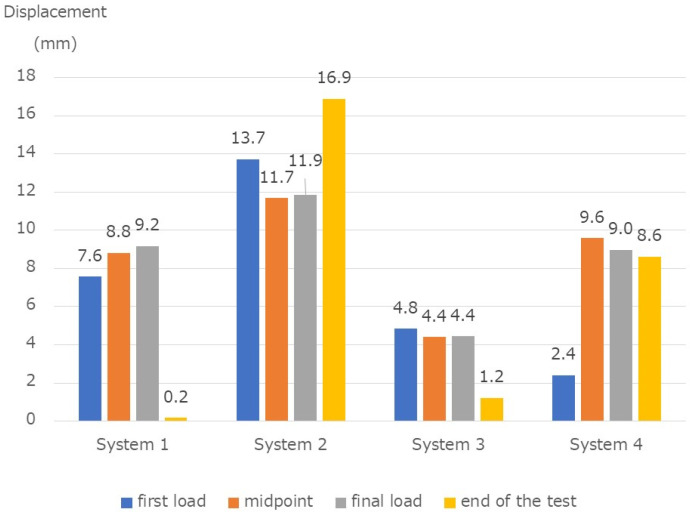
Fatigue test results. Displacement during one cycle at the first load, at the midpoint in the fatigue test, before the end of the test, and displacement after the end. The four systems showed statistically significant differences between them.

**Table 1 medicina-58-00565-t001:** Break test results.

	System 1	System 2	System 3	System 4
Yield value (N)	79.3	64.6	136.8	57.0
Stiffness (N/mm)	54.8	43.1	90.9	39.3
Failure	−	+	−	−

**Table 2 medicina-58-00565-t002:** Fatigue test results.

	Specimen No.	Number of Cycles	Broken Part	Displacement at First Load (mm)	Displacement at Midpoint (mm)	Displacement at Final Load (mm)	Displacement at the End (mm)
	1	35,273	Screw	8.6	9.5	10.6	−0.6
System 1	2	26,935	Screw	9.3	9.9	9.8	0.1
	3	340,972	Screw	4.8	7.1	7.1	−0.1
	4	54,781	Rod	14.3	12.7	12.6	21.1
System 2	5	23,818	Rod	13.7	12.0	12.1	14.4
	6	44,343	Screw	13.1	10.4	10.9	15.2
	7	2,000,000	-	4.9	4.3	4.3	1.1
System 3	8	651,166	Screw	4.9	4.6	4.7	1.2
	9	2,000,000	-	4.7	4.3	4.3	1.4
	10	161,819	Rod	2.3	9.5	9.7	8.0
System 4	11	136,557	Rod	2.3	9.4	9.7	8.3
	12	144,796	Rod	2.6	9.8	7.5	9.5

## Data Availability

Not applicable.

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
