# Peer review of "Mechanical Study of Various Pedicle Screw Systems including Percutaneous Pedicle Screw in Trauma Treatment"

_medicina, 2022, doi:10.3390/medicina58050565_

Round 1
Reviewer 1 Report
Although a good study design .... a large aspect fo fracture healing and stability after trauma remain bone quality and torsion strength which is not captured.
Even then I think the paper does provide some knowledge that can help surgeon on appropriate system to use on fracture fixation.
I would suggest to further describe certain fracture pattern that may warrant one system over another.
Author Response
Response to Reviewer 1 Comments
Point 1: Thank you for your review. As you pointed out, I didn't touch on bone quality issues. I will add the content related to this problem to the text.
Line 176
Response 1: Bone quality is very important in clinical practice. With good bone quality, loosening at the interface between the bone and the screw is unlikely to occur, so it is thought that it is possible to maintain the spine with greater force. In order to maintain the spine with a large correction force, it may be appropriate to select a thick rod system with high yield point and high stiffness. Using a thick rod will extend the life of the implant and may be advantageous even if it takes time to heal the fracture. On the other hand, if the bone quality is poor, there is a limit to the corrective maintenance of the spine by the spinal implant system (PPS), and it is considered that there is no merit in selecting an implant system with a high yield point and stiffness. When performing major surgical corrections for patients with poor bone quality, we consider that anterior reconstruction, vertebral plasty, and increasing number of screws should be performing to prevent large forces from concentrating on the spine and implants.
Point 2: I have described a specific fracture pattern. I added the following sentences to Line 167.
Response 2:
When there is a concern about initial failure (such as when AO classification A3 or higher with large angular deformation or when the large surgical correction is maintained only with the PPS system), it is better to avoid the conventional multiaxial screw. In the results of this study, it is considered that a system with a connecting mechanism with angular stability or a multiaxial screw system that has been improved so that initial failure is less likely to occur should be used.

Reviewer 2 Report
This is an interesting study on the biomechanical evaluation of different systems for both open and percutaneous spinal fixation. Although intriguing, results are really preliminary and the article suffers from from several issues. Most importantly, sample size is not clear as authors state that "one set of four spinal systems" was used (line 74) but later in Table 3 report 3 specimens per each system (moreover with significant intragroup differences). In the first case, absolute values in unrepeated experiments cannot be considered scientifically sound. If multiple trials on different specimens have been performed, the absence of a statistical analysis prevents any conclusion to be drawn. Authors should deeply revise this aspect before considering the possibility to publish their results.
Some minor comments:
- Moderate English language polishing is recommended. Improper use of syntax and lexicon can be noted in several points of the manuscript.
- Probably it would be better to describe PPS as "percutaneous pedicle screw placement" rather than "percutaneous pedicle screw", as it seem that authors only refer to one single screw and not to the tecnhique.
- Line 14: "Spine surgery using a percutaneous pedicle screw (PPS) is" should be changed to "The use of percutaneous pedicle screw placement (PPSP) in spine surgery is". Please change PPS to PPSP wherever applicable.
- Line 24: "Stiffness" should be changed to "Stiffness values"; "considered" should be changed to "reported".
- Line 26: Is there an extra comma in the fourth value? Please be consistent in reporting numbers (see line 118).
- Line 28: The noun "life" does not seem appropriate here. Please change it.
- Line 38: "The percutaneous pedicle screw (PPS) is a widely used minimal spinal surgical technique." should be changed to "Percutaneous pedicle screw placement (PPSP) is widely used in minimally invasive spine surgery".
- Line 40: These systems are intended for fracture fixation rather than reduction alone.
- Lines 61-72: Please organize the Systems in a bulleted list.
- Please enlarge Table 2 in order to improve its appearance.
- Table 3 should be edited as it cannot be read (the right part falls outside the page).
- Pictures of each described systems and their single component should be provided in order to increase readers' understanding.
- Limitations of the study should be moved from the Conclusion to the end of the Discussion.
Author Response
Response to Reviewer 2 Comments
Point 1: I would like to clarify the problem regarding the sample size. Destruction testing has done only one sample and the results are very limited. I will specify this. In the fatigue test, post hoc analysis was performed with an effect size of 0.6 and α error = 0.05, and the power was 0.25. It is impossible to draw a statistical conclusion. I will state that it is impossible to draw a conclusion.
Line 256
Response 1: The results of this experiment have not been able to draw statistical conclusions because the sample size is very small. Break tests have conducted only one sample of each system and the results are very limited. It is not a sample size that ensures sufficient power even in a fatigue test. It is difficult to draw conclusions based on the results of this study.
Point 2: Some minor comments:
I have corrected it as follows.
Response 2:
Line 14:
Spine surgery using a percutaneous pedicle screw (PPS) is widely implemented for spinal trauma.
→ Spine surgery using a percutaneous pedicle screw placement (PPSP) is widely implemented for spinal trauma.
Line 24:
Stiffness of 54.8 N/mm, 43.1 N/mm, 90.9 N/mm, 39.3 N/mm was considered for systems 1, 2, 3, and 4, respectively.
→ Stiffness value of 54.8 N/mm, 43.1 N/mm, 90.9 N/mm, 39.3 N/mm was reported for systems 1, 2, 3, and 4, respectively.
Line25:
The average number of load cycles in the fatigue test was 134393, 40980, 1550389, and 147,724 for systems 1 to 4.
→The average number of load cycles in the fatigue test was 134393, 40980, 1550389, and 147,724 for systems 1 to 4.
Line 118:
The average number of cycles at the end of the test was 134393, 40980, 1550389, and 147,724 for systems 1 to 4, respectively (Figure 6).
→ The average number of cycles at the end of the test was 134393, 40980, 1550389, and 147,724 for systems 1 to 4, respectively (Figure 6).
Line 38:
The percutaneous pedicle screw (PPS) is a widely used minimal spinal surgical technique.
→The percutaneous pedicle screw placement (PPSP) is a widely used minimal spinal surgical technique.
Lines 61-72: Please organize the Systems in a bulleted list.
- System 1 - open system for trauma, screw diameter: 6.2 mm, screw material: Ti-6AL-7Nb, rod diameter: 6.0 mm, rod material: titanium (Ti), connection mechanism: fixing system using clamps.
- System 2 - PPS system for trauma, screw diameter: 6.5 mm, screw material: Ti-6Al-4V, rod diameter: 6.0 mm, rod material: Ti-6Al-4V, connecting mechanism: fastening using a ball ring in the offset connector.
- System 3 - PPS system for trauma, screw diameter: 6.5 mm, screw material: screw shaft of Ti-6Al-4V, screw head of cobalt chrome alloy (not disclosed), rod diameter: 6.0 mm, rod material: cobalt-chromium alloy (not disclosed), connecting mechanism: using a sagittal adjusting screw (SAS).
- System 4 - general PPS system, screw diameter: 6.5 mm, rod material: Ti-6Al-4V, rod diameter: 5.5 mm, rod material: Ti-6Al-4V, connecting mechanism: a mechanism where the adapter inside the screw head holds down the screw head.
Table 2 has been enlarged to make it easier to see.
The right part of Tabel 3 is included in the page.
We will change it so that you can put a photo on it. The illustration of figure 2 will also be changed to a photo.
I changed the position of limitation from conclusion to the end of discussion

Round 2
Reviewer 2 Report
I appreciate that authors made a great effort in order to improve their manuscript. However, without the possibility to draw scientifically solid conclusions, the manuscript cannot be considered for publication.
I warmly invite the authors to increase the sample size, conduct an appropriate statistical analysis and then resubmit the paper.
Author Response
As you pointed out, the limit is that the sample size is small. We would like to increase the number of experiments, but it is difficult to continue the experiment by increasing the sample size in our experimental environment. Basic papers in the field of biomechanism (such as FEM and corpse experiments) often have limited sample sizes and small numbers. This experiment also has difficulty in procuring materials, and it is difficult to prepare a sufficient sample size. Could you give me some other solution?
Round 3
Reviewer 2 Report
I understand and sympathize with the authors' point of view. According to the Aims and Scope of Medicina, "the journal aims to advance knowledge related to problems in medicine [...], to disseminate research on global health, and to promote and foster prevention and treatment of diseases worldwide. MEDICINA publications cater to clinicians, diagnosticians and researchers, and serve as a forum to discuss the current status of health-related matters and their impact on a global and local scale." As you may have understood, the study is too preliminary in nature to match with this objectives and to reach a resonance that may have a significant impact. I suggest that you resubmit your manuscript to a more specialized Journal, where technical notes or short communications like yours may be accepted.